# An Updated Meta-Analysis on Long-Term Outcomes Following Hyperthermic Intraperitoneal Chemotherapy in Advanced Ovarian Cancer

**DOI:** 10.3390/cancers17091569

**Published:** 2025-05-05

**Authors:** Nadine El Kassis, Myriam Jerbaka, Rime Abou Chakra, Christopher El Hadi, Wissam Arab, Houssein El Hajj, Donal J. Brennan, David Atallah

**Affiliations:** 1Department of Gynecologic Surgery, Hotel-Dieu de France University Hospital, Beirut 16-6830, Lebanon; myriam.jerbaka@net.usj.edu.lb (M.J.); rim.bouchakra@net.usj.edu.lb (R.A.C.); nadine.kassis1@usj.edu.lb (N.E.K.); 2Faculty of Medicine, Department of Gynecology and Obstetrics, Saint-Joseph University, Beirut 17-5208, Lebanon; 3Faculty of Medicine, Lebanese American University, Beirut 13-5053, Lebanon; christopher.elhadi@laumcrh.com; 4Saint Mary’s Hospital, Manchester University NHS Foundation Trust, Manchester M33GZ, UK; wissam.arab@mft.nhs.uk; 5Institut Gustave Roussy, 94805 Villejuif, France; houssein.el-hajj@gustaveroussy.fr; 6School of Medicine, University College Dublin, D04 C1P1 Dublin, Ireland; donal.brennan@ucd.ie

**Keywords:** hyperthermic intraperitoneal chemotherapy, cytoreductive surgery, epithelial ovarian cancer, overall survival, progression-free survival, disease-free survival

## Abstract

Hyperthermic intraperitoneal chemotherapy has recently been broadly applied in the management of peritoneally disseminated cancers, including colorectal, appendiceal, and ovarian cancers. The evidence regarding its added value remains controversial. Vast differences still exist in clinical practice regarding the point at which this tool should be used along the ovarian disease journey (initial phase, before or after systemic chemotherapy, or following peritoneal relapse). This meta-analysis aims to obtain up-to-date evidence about the benefits of hyperthermic intraepithelial chemotherapy in different patient subgroups and to identify major complications. We found that hyperthermic intraperitoneal chemotherapy improves outcomes when used following neoadjuvant chemotherapy in primary cases and could lead to better outcomes when the open technique is used.

## 1. Introduction

Ovarian cancer is the leading cause of death among gynecological malignancies. Owing to the lack of screening tools and its non-specific presentation, the disease is usually not detected until in its advanced stages [1]. The standard of care for advanced ovarian cancer is cytoreductive surgery and platinum-based chemotherapy. Maintenance with antiangiogenic agents and poly (ADP-ribose) polymerase (PARP) inhibitors has recently been introduced and has improved long-term outcomes [2]. One of the most debatable management strategies in advanced ovarian cancer is hyperthermic intraperitoneal chemotherapy (HIPEC). Although the evidence is limited to retrospective studies and a few trials, the role of HIPEC in improving survival is increasingly recognized, particularly for advanced disease. However, consensus regarding the most effective drug, administration protocol, and postoperative therapy remains elusive [3]. HIPEC offers advantages over intraperitoneal chemotherapy alone, since hyperthermy allows for the direct involvement of cancer cells, enhanced chemotherapy cytotoxicity, inhibition of angiogenesis, improved protein denaturation, and tolerance. Controversies regarding HIPEC exist in both primary and recurrent EOC settings. Difficulties in including HIPEC as a standard of care stems from the variations and non-standardization of different studies. Therefore, this meta-analysis aims to assess CRS, with or without HIPEC, in terms of prevalence of relapse, OS, PFS, and postoperative complications.

## 2. Materials and Methods

### 2.1. Search Strategy and Study Selection

The search was carried out until December 2024 in the following sources: PubMed, EMBASE, Scopus, Web of Science, Google Scholar, Cochrane, Databases of Systematic Reviews, gray literature, ClinicalTrials.gov, and scientific meetings, particularly the 24th European Society of Gynecological Oncology (ESGO) 2023 Congress. This meta-analysis was registered in PROSPERO under registration number CRD42018102289. We proceeded with this investigation according to Preferred Reporting Items for Systematic Reviews and Meta-Analyses (PRISMA) guidelines [4,5]. We used the following keywords to search relevant articles: HIPEC (“hyperthermic intraperitoneal chemotherapy” or “HIPEC”), ovary (“ovarian” or “ovary”), and cancer (“cancer” or “carcinoma” or “neoplasm” or “malignancy” or “tumor”). A manual search of the reference lists of included studies was performed to identify additional relevant articles. Efficient study selection was ensured by the use of the Rayyan application. In addition, a review of existing systematic reviews and meta-analyses relating to HIPEC in EOC was carried out.

We included studies that met the following criteria: (1) comparative analysis including patients diagnosed with advanced primary or recurrent EOC and treated with CRS and HIPEC vs. CRS alone; (2) case-control, cohort studies, nonrandomized, or randomized controlled trials (RCTs) on humans published in English, French, or German.

We excluded case reports, case series, animal experiments, phase I/II trials, non-comparative studies with a single HIPEC arm, and systematic reviews. Phase I and II trials for a drug are often excluded because they lack randomization or could be single-arm studies, leading to biased effect estimates.

Two independent reviewers (M.J., R.A.C.) screened the title and abstract of each article for relevance (Figure 1). The full-text articles were then retrieved for further detailed review for confirmation of study inclusion. Any disagreement was resolved by consensus.

### 2.2. Data Extraction

The data collected included: first author, year of study and of publication, study population, number of patients, experimental and control arm with therapy details, HIPEC technique, median follow-up, median OS and PFS reported in months, and rates of complications and their grade (according to Clavien–Dindo Classification).

### 2.3. Quality Assessment and Statistical Analysis

We evaluated the methodological quality of RCTs based on the Cochrane Collaboration (ROB) [6,7], and case-control and cohort studies based on the Newcastle–Ottawa Scale (NOS) [8]. Hazard ratios (HRs) were extracted for OS and PFS, alongside their 95% confidence intervals (CIs), when available. To calculate the relative risk ratios (RRs), we used the rates of grade III and IV postoperative complications in two groups, with or without HIPEC. When data were not available, HRs were calculated using the recommended formulas. Survival curves were digitized if data were not available, taking censoring into account [9,10]. Analysis was performed using R version 4.3.1 [11]. Forest plots were used to illustrate study results and overall estimates.

Statistical heterogeneity was assessed using I^2^, with *p* > 0.10 indicating the absence of significant heterogeneity, and a fixed-effects model; otherwise, a random-effects model was used. Subgroup analyses were performed based on study design. Univariate meta-regression explored the variability of the treatment effect. PDS, SCR, and IDS groups were formed and analyzed in the same manner. Funnel plots were used to assess publication bias, with Egger’s test examining bias within studies.

## 3. Results

### 3.1. Characteristics of Included Studies

Our research methodology is depicted in Figure 1 and Figure 2. Nineteen articles were included in our meta-analysis: six RCTs, four case-control studies and nine cohort studies. Eligible studies were conducted between 1991 and 2024 in The Netherlands, South Korea, Spain, Italy, Greece, France, Belgium, Canada, China, Brazil, Switzerland, and the USA. A total of 2666 patients were analyzed, with 1227 undergoing CRS alone vs. 1439 undergoing CRS with HIPEC. Details of the included studies are illustrated in Table 1, and PICO criteria for this meta-analysis are shown in Table 2.

During full-text assessment, we excluded some studies due to the following reasons: the study by Wu [12] was excluded because the control group received IV NACT alone, without surgery. The studies by Marrelli et al. [13] and Batista et al. [14] were excluded for being phase II trials. The study conducted by Cascales-Campos [15] did not provide clear results, making it unsuitable for inclusion. This study was initially designed to include 126 patients, but ultimately included 71 patients, and the control arm inlcuded a higher proportion of Stage IV EOC (17% vs. 6%). Also, the information regarding histological subtypes was unavailable, and it is unclear whether patients continued treatment with bevacizumab or PARP inhibitors.

**Table 1 cancers-17-01569-t001:** Details of included studies.

Author (Study Type/Interval/Country)	Surgery Type	Number of Patients (Control/HIPEC arm)	Treatment	HIPEC Scheme	HIPEC Technique	HIPEC T(°C), Duration (min)	Survival Analysis
Aronson et al. [16] (RCT/2007–2016/Netherlands)	IDS	123/122	NACT + CRS ± HIPEC	CDDP 100 mg/m^2^	Open	40–42, 90’	OS/PFS
Baiocci G. et al. [17] (Cohort/2000–2014/Brazil)	SCR	50/29	NACT + CRS ± HIPEC	MMC 10 mg/m^2^ + CDDP 50 mg/m^2^ or CDDP 50 mg/m^2^ + DXR or CDDP 50 mg/m^2^ or OX	Closed	41–42, <90’	OS/DFS
Campos et al. [18] (RCT/2012–2019/Spain)	PDS, IDS and SCR	23/32	NACT ± CRS ± HIPEC	PTX 175 mg/m^2^	Closed	42–43, 60’	OS/RFS/AE
Ceresoli et al. [19] (CC/2010–2016/Italy)	IDS	28/28	NACT + CRS ± HIPEC	CDDP 100 mg/m^2^ + PTX 175 mg/m^2^	Open	41.5, 90’	OS/DFS/AE
Classe et al. [20] (RCT/2011–2021/France)	SCR	208/207	NACT + CRS ± HIPEC	CDDP 75 mg/m^2^	Open or closed	41 ± 1, 60’	OS/PFS/AE
Fagotti et al. [21] (CC/2005–2009/Italy)	SCR	13/30	CRS ± HIPEC	OX 460 mg/m^2^	Closed	41.5, 30’	OS/PFS
Kim et al. [22] (CC/1991–2004/South Korea)	PDS	24/19	CRS ± HIPEC	PTX 175 mg/m^2^	Open	43–44, 90’	PFS/OS
Le Brun et al. [23] (CC/1997–2011/France)	SCR	19/23	NACT+ CRS ± HIPEC	CDDP or OX or MMC	NA	42, Cis 60’, OX 30’, MMC 30’	OS
Lee et al., 2022 [24] (Cohort/2015–2019/South Korea)	IDS	80/43	NACT + CRS ± HIPEC	CDDP 100 mg/m^2^ or PTX 175 mg/m^2^	Closed	42, 90’	PFS/OS/AE
Lee et al., 2023 [25] (Cohort/2017–2022/South Korea)	IDS	87/109	CRS ± HIPEC	CDDP or PTX	Open and closed	42, 90’	OS/PFS/AE
Lei et al. [26] (Cohort/2010–2017/China)	PDS	159/425	CRS ± HIPEC	CDDP 50 mg/m^2^	Closed	43, 60’	OS/AE
Lim et al. [27] (RCT/2010–2016/South Korea)	PDS or IDS	92/92	NACT+ CRS ± HIPEC	CDDP 75 mg/m^2^	Closed	41.5, 90’	OS/PFS/AE
Marocco et al. [28] (Cohort/1995–2012/Italy)	SCR	11/19	NACT + CRS ± HIPEC	CDDP 100 mg/m^2^ + DXR 15.2 mg	Semi-closed	41.5, 60’	PFS/OS
Mendivil et al. [29] (Cohort/2008–2014/USA)	PDS	69/69	CRS ± HIPEC	CB AUC10	Closed	41.5, 90’	PFS/OS
Muñoz-Casares et al. [30] (Cohort/1997–2004/Spain)	SCR	12/14	CRS ± HIPEC	PTX 60 mg/m^2^	Open	41–43, 60’	OS/AE
Spiliotis et al. (PR) [31] (RCT/2006–2013/Greece)	SCR	24/22	CRS ± HIPEC	DXR 35 mg/m^2^ + PTX 175 mg/m^2^	Open and closed	42.5, 60’	RFS/OS
Spiliotis et al. (PS) [31] (RCT/2006–2013/Greece)	SCR	36/38	CRS ± HIPEC	CDDP 100 mg/m^2^ + PTX 175 mg/m^2^	Open and closed	42.5, 60’	RFS/OS
Van Driel et al. [32] (RCT/2007–2016/Netherlands)	IDS	123/122	NACT + CRS ± HIPEC	CDDP 100 mg/m^2^	Open	40, 120’	OS/PFS/AE
Warschkow et al. (PDS) [33] (Cohort/1991–2006/Switzerland)	PDS	43/10	CRS ± HIPEC	CDDP 50 mg/m^2^	NA	42, 90’	OS/AE
Warschkow et al. (SCR) [33] (Cohort/1991–2006/Switzerland)	SCR	47/11	CRS ± HIPEC	CDDP 50 mg/m^2^	NA	42, 90’	OS/AE
Zhang et al. [34] (Cohort/2004–2019/China)	PDS	53/80	CRS ± HIPEC	CDDP 120 mg + MMC 30 mg DTX/PTX 120 mg + CDDP 30 mg	Open	43, 60’	OS/PFS
Zivanovic et al. [35] (RCT/2014–2016/USA)	SCR	49/49	CRS ± HIPEC	CB 800 mg/m^2^	Closed	41–43, 90’	OS/PFS

Abbreviations: AE: adverse events, AUC: area under the curve, CB: carboplatin, CC: case-control study, CDDP: cisplatin, CRS: cytoreductive surgery, DFS: disease-free survival, DTX: Docetaxel, DXR: doxorubicin, HIPEC: hyperthermic intraperitoneal chemotherapy, IDS: interval debulking surgery, MMC; Mitomycin, NACT: neoadjuvant chemotherapy, OS: overall survival, OX: oxaliplatin, PR: platinum-resistant, PDS: primary debulking surgery, PS: platinum-susceptible, PFS: progression-free survival, PTX: paclitaxel, RCT: randomized controlled trial, RFS: recurrence-free survival, SCR: secondary cytoreductive surgery, and T: temperature.

**Table 2 cancers-17-01569-t002:** PICO criteria.

Population	Women with primary epithelial ovarian carcinoma in FIGO Stages III or IVa
	Women with recurrent epithelial ovarian carcinoma
Intervention	With or without neoadjuvant chemotherapy Cytoreductive surgery ± HIPEC ± adjuvant chemotherapy
Comparison	CRS ± chemotherapy (neoadjuvant ± adjuvant) vs. CRS + HIPEC ± chemotherapy (neoadjuvant ± adjuvant)
Outcome	Overall survival (OS)
	Progression-free survival (PFS)
	Complications
Study design	Randomized controlled trial (RCT)
	Case-control (CC)
	Cohort

### 3.2. Quality Assessment

Four case-control studies scored between 7 and 8, and nine cohort studies scored 7, indicating moderate-to-high quality. The six RCTs included show a moderate risk of bias and provide moderate- to high-quality evidence.

### 3.3. Risk of Publication Bias

Funnel plots for both PFS and OS were symmetrical, and Egger’s test results for OS and PFS (*p* = 0.317 and *p* = 0.8, respectively) were insignificant, implying no publication bias in this meta-analysis (Figure 3).

### 3.4. Statistical Outcomes According to Study Design

The effect size beta (β) is the logarithm of the HR: a decrease in effect size corresponds to an improvement in survival outcomes. For OS outcomes, studies with higher levels of evidence, such as RCTs and cohort studies, were less likely to demonstrate the benefits of HIPEC (*p* = 0.0072). The duration of the HIPEC procedure also influenced the outcomes. Longer HIPEC durations were associated with lower OS and PFS in secondary CRS, with OS and PFS reporting β = 0.01 (0–0.02). No significant variables were found to modify PFS in any of the studies.

### 3.5. Primary Outcome (Survival Analysis)

The OS results are summarized in Figure 4. The random effect model revealed that HIPEC increases OS, lowering the risk of death to 35.5% for patients undergoing IDS and 48.5% for those undergoing SCR (HR = 0.65, 95% CI: 0.58, 0.72; *p* < 0.0001). As for PFS, the random effects model was also required, showing a significant improvement in pooled PFS (HR = 0.66, 95% CI: 0.55–0.79) The OS results are summarized in Figure 4 and the PFS results in Figure 5.

#### 3.5.1. HIPEC in Primary CRS

Six studies including 1135 patients assessed the use of HIPEC in PDS [22,26,27,29,33,34]. There were 440 patients in the control group and 695 in experimental group. No statistically significant difference in OS was found, with an HR of 0.54 (95% CI: 0.281, 1.052). On the other hand, HIPEC during PDS provided improvement in PFS, with an HR of 0.473 (95% CI: 0.334, 0.669).

Concerning the HIPEC regimen, paclitaxel monotherapy, as well as cisplatin monotherapy, both appeared to provide survival benefits in patients undergoing PDS (*p* = 0.026 and *p* = 0.011). For PDS, studies including patients on a paclitaxel-only regimen [22] reported superior results (β = −1.64, 95% CI: −3.03, −0.24) compared to those using a cisplatin-only regimen (β = 1.26, 95% CI: 0.36, 2.15) [26,27,33]. In PDS with HIPEC, higher intraperitoneal temperatures (β = −0.79, 95% CI: −1.42, −0.16) and the use of open techniques (*p* = 0.0475) were correlated with better OS. According to age, older patients (more than 65 years old) appear to benefit the most from the use of HIPEC during PDS with the open technique (β = −0.15, 95% CI: −0.27, −0.04). Moreover, they showed better HR results (β = −0.13; 95% CI: −0.25, −0.01, for every year older than 65).

#### 3.5.2. HIPEC in IDS

Five studies investigated the use of HIPEC in IDS [19,24,25,27,32] and included a total of 804 patients (410 in the control group and 394 in experimental group). Patients who underwent IDS with HIPEC showed significant improvement in OS (HR = 0.632, 95% CI:0.524, 0.761) and PFS (HR = 0.631, 95% CI: 0.543, 0.734). Overall, when IDS was combined with HIPEC, the risk of recurrence or death was lowered by 35.5%.

Three of the studies on HIPEC and IDS [16,24,25,32] used cisplatin monotherapy, while one study employed cisplatin-combined or paclitaxel-combined regimens [19] and another used a paclitaxel monotherapy regimen [25]. Among the 394 patients in the experimental group, 204 were treated with an open HIPEC and 189 with the closed technique. Overall, the open technique showed better results (HR = 0.49, 95% CI: 0.34, 0.73) compared to those of the closed technique (HR = 0.84, 95% CI: 0.63, 1.13).

#### 3.5.3. HIPEC in Relapse Surgery

Nine studies investigated HIPEC during SCR, with a total of 911 patients (442 with HIPEC and 469 without HIPEC) [17,20,21,23,28,30,31,33,35]. When HIPEC is applied in SCR for patients with recurrent EOC, better OS was obtained (HR = 0.66, 95% CI:0.49, 0.89). Conversely, four of these studies investigated the PFS and did not demonstrate a statistically significant improvement (HR = 0.83, 95% CI: 0.53, 1.31) [20,21,28,35]. The CHIPOR trial showed improvement in PFS, with a median of 10.2 months with HIPEC and 9.5 months without HIPEC [20]. The stratified HR was 0.79 (95% CI: 0.63, 0.99), suggesting a statistically significant reduction in risk of progression. Though analysis of peritoneal PFS and time to first subsequent therapy also favored HIPEC, but this result was less clear for extraperitoneal PFS [20]. 

With regards to recurrent EOC and the HIPEC regimens during SCR, the oxaliplatin-only regimen showed better prognosis (β = −1.15, 95% CI: −1.96, −0.35) [21]. When doxorubicin was used in combination with another agent, the OS was worse (β = 1.22, 95% CI: 0.06, 2.38) [17,28,31] than when doxorubicin was omitted from the regimen (β = −0.83, 95% CI: −1.57, −0.10) [21,23,30,31,33,35]. 

As for SCR, each unit of increase in the peritoneal cancer index (PCI) produced a significant improvement in the HR (β = −0.14, 95% CI: −0.26, −0.03) when HIPEC was added to surgical treatment.

#### 3.5.4. HIPEC Regimen and Technique

For all studies combined, those including patients on the paclitaxel-only regimen during HIPEC showed better outcomes (β = −0.67, 95% CI: −1.33, −0.02) [22,30]. Across different surgical contexts (primary, interval, or secondary cytoreductive surgeries), HIPEC protocols involving paclitaxel combined with other cytotoxic agents were associated with improved OS (β = −0.59, 95% CI: −1.04, −0.13). Oxaliplatin as an HIPEC regimen was associated with better prognosis compared with the results for regimens without oxaliplatin (OS = 0.26, 95% CI: 0.11, 0.64 vs. OS = 0.69, 95% CI: 0.53, 0.89). Regarding the employed technique, open HIPEC had a better effect on survival (HR = 0.49, 95% CI: 0.34, 0.73) compared to the closed technique (HR = 0.84, 95% CI: 0.63, 1.13).

#### 3.5.5. Complications

The incidence of procedure-related grade III and IV complications was significantly higher in the HIPEC with CRS group, with a pooled RR of 1.40 (1.08–1.81), suggesting a 40% higher risk within one month of the HIPEC procedure compared to that for CRS alone (Figure 6). Severe renal intraprocedural adverse events were rare, with only three reported cases and a single instance of hydronephrosis as a grade III complication. Amifostine was effective in reducing serum creatinine levels and acute kidney injury in the HIPEC group. The HIPEC group had a 34.8% rate of severe toxicity within thirty days. This group also experienced a higher prevalence of grade I and II complications. The most common complications included anemia, thrombocytopenia, neutropenia, ileus, and nausea. Severe grade III or IV events occurred in the HIPEC group, with electrolyte disturbances being the most common. However, there were no intraprocedural adverse events or deaths within 30 days. Kim et al. noted transient hepatitis in 11% of the HIPEC-paclitaxel group, which resolved without intervention [22]. Pleural effusion and hydronephrosis were common grade III adverse events in the IDS with HIPEC group. Major surgical complications in the HIPEC group included bleeding, infection, and leaks.

Overall, no intraprocedural adverse events or HIPEC-related deaths were reported, maintaining a 0% 30-day mortality rate.

## 4. Discussion

Ovarian cancer remains the most fatal gynecological malignancy, with no established screening tool at this time. Diagnosis is often made at advanced stages, and the long-term survival rates remain low, despite all the advancements in its management. Interest in HIPEC continues to grow, raising new questions about its optimal application timing (interval vs. upfront HIPEC), as well as its safety and place among the plethora of newly introduced targeted therapies [36,37]. Many experts, as well as oncology societies, still consider HIPEC as experimental and therefore, that it should be used with caution, considering its potential complications and unclear benefits. Our purpose was to identify the specific subgroups that are likely to benefit from HIPEC, recognizing subgroups for which its application is unnecessary and possibly harmful.

Theoretically, it is well known that hyperthermia increases the drug’s penetration into tumor tissues and enhances its cytotoxic effects. Hyperthermia itself also allows for better immune reactions to cancerous cells by allowing the accumulation of heat shock proteins at the surface of the dying tumor cells.

In the present meta-analysis, we identified the patient groups and cancer characteristics most likely to benefit from HIPEC’s survival-enhancing potential. Our review revealed that the open HIPEC technique used during IDS provides improved outcomes in terms of OS and PFS compared to those for IDS alone or IDS with HIPEC using closed and semi-closed techniques (HR = 0.49, 95% CI: 0.34, 0.73). The advantages of the open technique include an even distribution of the chemoperfusate in the abdominal cavity and the possibility to perform anastomoses after HIPEC, lessening the risk to the integrity of healthy cells and the perforation of the connection. However, its disadvantages include heat dissipation and the risk of personnel exposure to chemotherapeutic agents. The closed technique has been associated with the uneven distribution of the perfusate in the peritoneal cavity, but it prevents the exposure of the surgical team. HIPEC had the greatest impact during IDS, confirming findings from two recent meta-analyses [38,39]. The reasons why HIPEC seems to produce better results after systemic chemotherapy are unclear. While the goal of every CRS is to remove all visible tumors, chemo-resistant tumors are not visually distinct and can only be resected when macroscopically visible. Proponents of HIPEC argue that administering NAC before CRS may selectively reduce chemo-sensitive tumors, thereby increasing the exposure of residual chemo-resistant tumors or cancer stem cells to HIPEC during surgery. On the other hand, proponents of primary cytoreduction consider that improved survival derived from HIPEC during IDS is due to the fact that IDS is inferior to PDS. HIPEC during IDS could be compensating for the lower cytoreduction, since the visualization of tumor-affected areas might be impaired following NACT [36].

Our findings in terms of better survival rates in IDS groups meet those of other meta-analysis published previously. Della Corte, L., et al. [38] published a meta-analysis in 2023 claiming that the combination of HIPEC with CRS prolongs OS and PFS in advanced EOC after NACT, with acceptable morbidity. Nevertheless, these meta-analyses included only two RCTs, while we included five, indicating that our meta-analysis provides a strong update. The meta-analysis conducted in 2020 by Cianci, S., [40] suggests that HIPEC during SCR may improve the survival rate, even after five years of follow-up.

The use of cisplatin as chemotherapy in HIPEC is recommended by the National Comprehensive Cancer Network (NCCN) guidelines at a dose of 100 mg/m^2^ [41]. Our results showed improved outcomes in IDS with cisplatin or paclitaxel monotherapy, while oxaliplatin might be effective in SCR. Moreover, the pooled outcomes in our meta-analysis were in favor of paclitaxel. For all studies combined, those including patients on a paclitaxel-only regimen showed improved outcomes (β = −0.67, −1.33 to −0.02). During HIPEC, paclitaxel is administered directly into the peritoneal cavity at a concentration of 50–100 mg/m^2^. The solution is heated to around 41–42 °C and circulated for approximately 90 min. A randomized phase III clinical trial, HIPECOVA [18], evaluated the effectiveness of paclitaxel in HIPEC following surgical cytoreduction for ovarian peritoneal metastases. We could not include this trial because the results are not stratified in subgroups (primary, interval, or secondary debulking surgery). The trial showed improved 5-year OS and 3-year RFS rates in the HIPEC group.

As for the safety of HIPEC, our meta-analysis found that the risk of morbidity was higher compared to that for the surgery-alone group, with a higher incidence in the HIPEC group, i.e., a pooled RR of 1.40 (95% CI: 1.08–1.81), suggesting a 40% higher risk of grade III-IV complications within one month of the HIPEC procedure. The most common complication is electrolyte disturbance. Multiple studies reported that the rate of postoperative complications after HIPEC in colorectal cancer was high and could affect the survival outcome.

Our results concerning complications are not strong because the number of articles that report complications is small. In the literature, many reviews identified a high rate of HIPEC-related complications. In a recent systematic review by Navarro Santana, B., et al. [42], complications were evaluated and compared over two time periods, i.e., 2004–2013 and 2014–2022, showing no changes, except in regards to hemoperitoneum, which was lower in the second period. The mortality rate was 0–7%, which is similar to the result in our meta-analysis. Multiple studies published that the rate of postoperative complications after HIPEC in colorectal cancer was high and could affect the survival outcome. One of the examples is the metanalysis conducted by Narasimha et al. [43], that included 717 patients undergoing surgery with HIPEC. The study showed that the presence of grade III and IV morbidity was associated with worse OS (HR, 1.59, 95% CI, 1.17–2.16; *p* = 0.003). 

Therefore, we suggest that from a patient safety perspective, HIPEC should be offered to patients in the context of a clinical trial, whether for primary or recurrent cases. The benefits and inconveniences should be discussed (CRS alone vs. HIPEC) by informing patients about the types and frequencies of complications of this technique.

On the other hand, mitigating the complications in the practice of HIPEC for ovarian cancer involves adhering to standardized perioperative care protocols and implementing strategies aimed at reducing severe adverse events. We should practice careful patient selection and preoperative assessment, as well as adherence to standardized surgical protocols and HIPEC procedures. Experienced multidisciplinary teams should be involved, and perioperative management should include balanced fluid management, temperature and perfusion monitoring, the use of pre- and intra-operative sodium thiosulfate, postoperative care and monitoring (enhanced recovery after surgery (ERAS) and vigilant monitoring for complications), and continuous auditing and quality improvement.

In this meta-analysis, the shortage of RCTs was likely to increase the risk of bias. Secondly, most of the studies were observational, which might compromise the results. Thirdly, confounding factors, such as EOC platinum-resistance of the disease, pathological classification and stage of the disease, the type of chemotherapy used, the use of PARPi and anti-angiogenic agents after HIPEC, as well as the completeness of cytoreduction need to be further stratified to determine the most suitable candidates for HIPEC. Other confounding factors include the age of the patients and their performance status, other comorbidities, different anesthetic protocols, the heterogeneity of follow-up length, and differences in intervals from last chemotherapy in a neoadjuvant setting. Fourth, we could not expand the comparison to the adjuvant chemotherapy used and the use of bevacizumab and PARP (Poly ADP-Ribose Polymerase) inhibitors. Therefore, future research should focus on the molecular benefits of HIPEC in specific populations, the role of PARP inhibitors and bevacizumab in survival outcomes in patients receiving HIPEC, and the more comprehensive assessment of long-term outcomes and quality of life. Finally, a new economic evaluation and the collection of global data for the practicality and accessibility of HIPEC are needed. Meanwhile we are looking forward to the results of OVHIPEC-2 trial, that will include 538 patients with newly diagnosed FIGO stage III epithelial ovarian, fallopian tube, or primary peritoneal cancer and other ongoing RCTs, which will help clarify the benefits of HIPEC in EOC and guide future treatment protocols [44].

## 5. Conclusions

This updated meta-analysis shows that the use of HIPEC during CRS improves OS and PFS in patients undergoing interval debulking surgery for advanced epithelial ovarian cancer. The open HIPEC technique yielded superior results compared to those for the closed technique. However, the use of HIPEC should be approached with caution due to the non-negligible number of grade III and IV complications compared to those for surgery alone. Further insights are expected from the results of the OVHIPEC-2 trial, which will include 538 patients with newly diagnosed FIGO stage III epithelial ovarian, fallopian tube, or primary peritoneal cancer and other ongoing RCTs, therefore offering stronger evidence to guide clinical decision making.

## Figures and Tables

**Figure 1 cancers-17-01569-f001:**
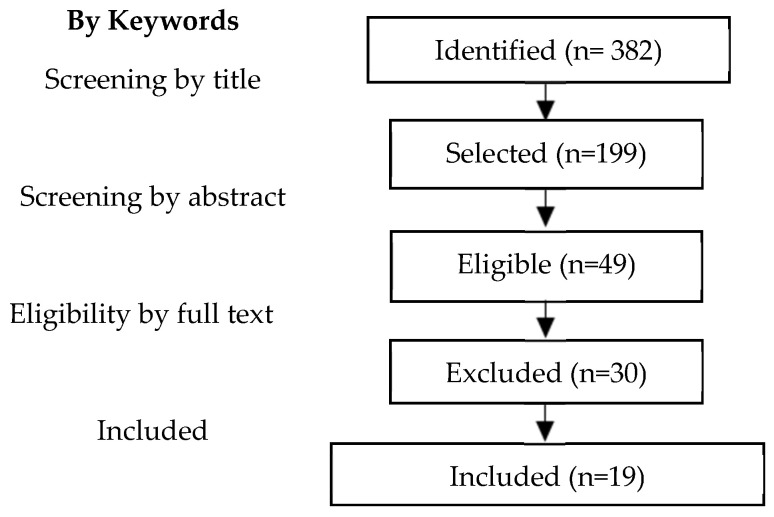
Study selection and research strategy.

**Figure 2 cancers-17-01569-f002:**
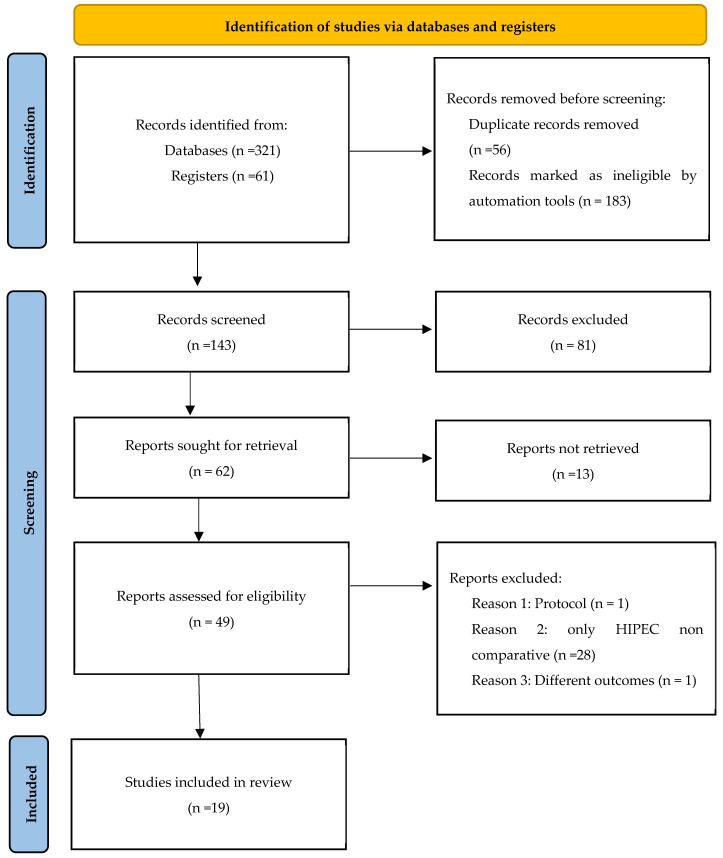
Flowchart illustrating the systematic review and meta-analysis process conducted in this study, adapted from the PRISMA 2020 statement, providing updated guidelines for systematic review reporting. (Source: Page, M.J., McKenzie, J.E., Bossuyt, P.M., Boutron, I., Hoffmann, T.C., Mulrow, C.D., et al. BMJ 2021;372:n71. doi: 10.1136/bmj.n71. [5] For additional details, refer to: http://www.prisma-statement.org/ [4]) accessed on 30 July 2024.

**Figure 3 cancers-17-01569-f003:**
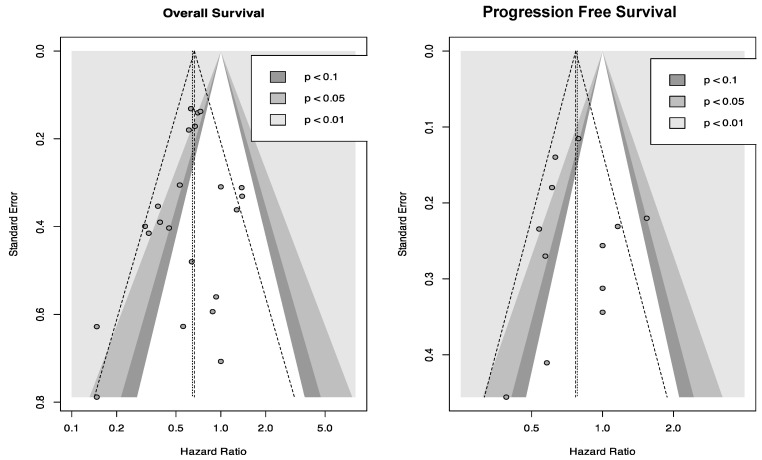
Funnel plots assessing publication bias among the included studies. (**Left plot**) Funnel plot for OS. (**Right plot**) Funnel plot for PFS. The vertical line represents the pooled HR effect size, and the diagonal lines indicate the expected 90, 95 and 99% confidence intervals.

**Figure 4 cancers-17-01569-f004:**
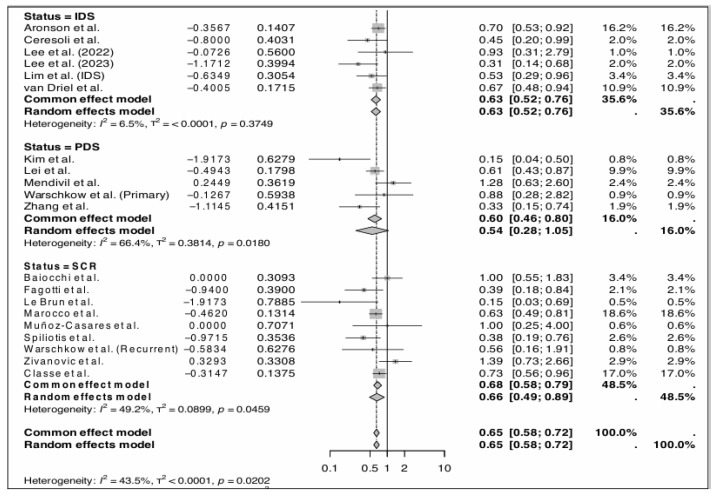
Forest plot for the pooled overall survival hazard ratio, featuring primary and recurrent ovarian cancer subgroups [16,17,19,20,21,22,23,24,25,26,27,28,29,30,31,32,33,34,35].

**Figure 5 cancers-17-01569-f005:**
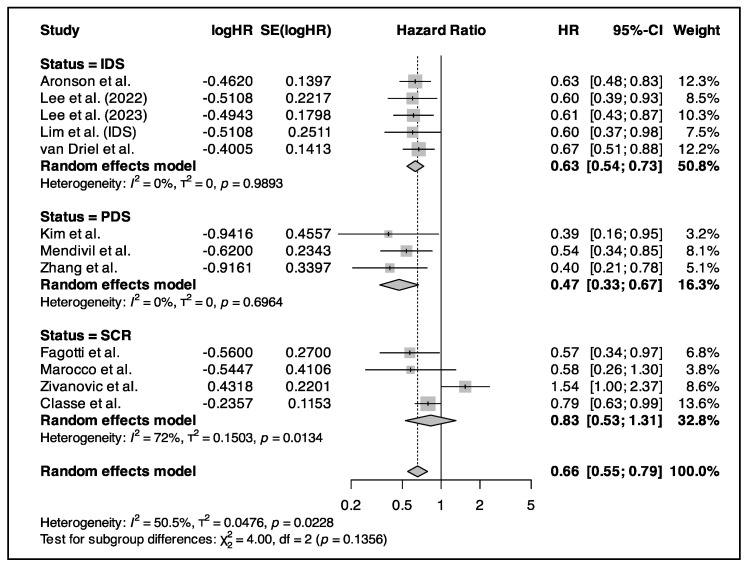
Forest plot for the pooled progression-free survival hazard ratio, featuring primary and recurrent ovarian cancer subgroups [16,20,21,22,24,25,27,28,29,32,34,35].

**Figure 6 cancers-17-01569-f006:**
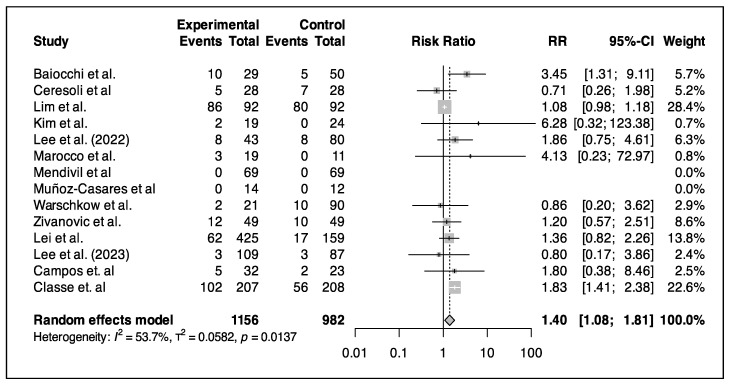
Forest plot of relative risks reported by the studies [17,18,19,20,22,24,25,26,27,28,29,30,33,35].

## Data Availability

Pooled data and statistical sheets are available upon request.

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
