# Peer review of "An Updated Meta-Analysis on Long-Term Outcomes Following Hyperthermic Intraperitoneal Chemotherapy in Advanced Ovarian Cancer"

_cancers, 2025, doi:10.3390/cancers17091569_

Round 1

Reviewer 1 Report

Comments and Suggestions for Authors

I was very pleased to read the meta-analysis by respected Nadine El Kassis et al. Updated meta-analysis is devoted to long-term outcomes following hyperthermic intraperitoneal chemotherapy in advanced ovarian cancer. The strengths of the review are the richness of the material: 2666 patients, 6 RCTs, 9 cohorts, several case-control studies, a large range of studied outcomes. Another strength of this meta-analysis is its great practical significance.

The relevance of the topic is due, firstly, to the importance of cancer therapy. Hyperthermic intraperitoneal chemotherapy is an innovative and promising method of therapy of metastases, and its use can significantly improve the effectiveness of anticancer treatment.

The meta-analysis is clearly structured, the main part consists of five main sections. They provide a description of the studies, quality assessment, risk of publication bias, statistical outcomes, survival analysis, and complications. The methods and approaches are fully consistent with the stated goal, the meta-analysis is quite complete. There are several meta-analyses published on hyperthermic intraperitoneal chemotherapy, but in recent years they have been devoted to gastrointestinal diseases. The previous meta-analysis on hyperthermic intraperitoneal chemotherapy in ovarian cancer is dated 2022 (10.1016/j.ygyno.2022.10.010). A lot of material has been accumulated in recent years, so the authors' meta-analysis offers a valuable addition to previous works. Moreover, it deals with long-term outcomes.

The study includes 44 references, 28 of which are from the last five years. Thus, the authors' study contains up-to-date information.

The conclusions made by the authors are correct and supported by references.

The review contains six original figures illustrating the search strategy, meta-analysis flow chart, funnel plots for the studies, forest plots, as well as two tables summarizing the details of included studies and PICO criteria.

I highly appreciate the stady of the authors and believe that it can be published in present form.

Reviewer 2 Report

Comments and Suggestions for Authors

This meta-analysis of hyperthermic intraperitoneal chemotherapy (HIPEC) for the treatment of advanced ovarian cancer is comprehensive and detailed, but contains several significant inconsistencies and methodological flaws.

First, there is a significant inconsistency in the terminology used throughout the article, particularly in the distinction between event-free survival (EFS), progression-free survival (PFS) and disease-free survival (DFS). In the abstract, these terms are used interchangeably without a clear distinction, which may mislead the reader as to exactly what outcomes are being reported and compared.

In addition, the criteria for inclusion in the trials are unclear. Although the authors mention the exclusion of phase I/II trials and single-arm trials, they do not clearly justify why these exclusions are necessary or how their absence might affect the results. 

The article also fails to adequately consider potential confounding factors, such as the different lengths of follow-up and different baseline characteristics of patients in the different trials. 

Furthermore, despite acknowledging the higher complication rate associated with HIPEC, the discussion does not adequately address this important clinical issue. The authors downplay the significance of the increased number of Grade III-IV complications without providing clear guidelines on managing or mitigating this risk in clinical practice.

Finally, the article concludes by mentioning the ongoing OVHIPEC-2 trial, suggesting future clarity, but lacking a robust discussion of how current limitations may affect clinical decision-making or future research priorities.

In conclusion, although the article provides valuable insights, particularly regarding the potential benefits of HIPEC in interval debulking surgery, its methodological weaknesses and inconsistencies should be improved.

Reviewer 3 Report

Comments and Suggestions for Authors

Presented manuscript is a well-structured and designated research that describes an updated meta-analysis on long-term outcomes following hyperthermic intraperitoneal chemotherapy in advanced ovarian cancer.

Since ovarian cancer is the first cause of death among gynecological malignancies and there is an urgent need in diagnostic and therapy methods, it is an actual research.

The manuscript contains number of shortcomings and typing errors (lines 84, 93 etc) that need to be corrected.

Figure 1, data selection: mismatch between number of eligible (40), excluded (30) and included (19) data: 19 plus 30 not equal 48.

Table 1. Is it possible to mention all 19 included studies? Please recheck and correct data: Surgery column is presented for first 5 records only. This resulted to data shift till the table end.

Round 2

Reviewer 2 Report

Comments and Suggestions for Authors

The current version of the manuscript is suitable for publication in Cancers.